# Iris-Claw Intraocular Lens Implantation in Pediatric Population: Indications, Outcomes, and a Comparison with Adult Population

**DOI:** 10.3390/jcm14041135

**Published:** 2025-02-10

**Authors:** Rami Al-Dwairi, Hisham M. Jammal, Mohammad Al Qudah, Hamad Alazmi, Saad Almutairi, Abdelwahab Aleshawi

**Affiliations:** 1Department of Special Surgery, Division of Ophthalmology, Faculty of Medicine, Jordan University of Science & Technology, Irbid 22110, Jordan; hmjammal@just.edu.jo (H.M.J.); abdelwahhabjamal@yahoo.com (A.A.); 2Department of Ophthalmology, Ministry of Health, Kuwait City 12009, Kuwait

**Keywords:** ectopia lentis, artisan, trauma, glaucoma

## Abstract

**Objective**: The utilization of iris-claw intraocular lens (IOL) in adult cases of capsular insufficiency has been investigated. However, the use of iris-claw IOL in the pediatric population is still under investigation. In this study, we evaluated the overall practice of iris-claw IOL implantation in pediatrics and compared the visual outcomes and postoperative complication rates between children and adults. **Methods:** Retrospectively, we examined the practice, indications, and outcomes of pediatric patients who underwent iris-claw “Artisan^®^” intraocular lens implantation (IOL). All patients who underwent iris-claw IOL implantation for any causative indications were enrolled in the study. The outcome was compared by visual acuity and postoperative complications. Furthermore, the outcomes in pediatrics were compared to another adult group. **Results:** In this study, 41 eyes of 34 pediatric patients were included with a mean age of 11.1 years and a mean follow-up period of 24.5 months. Trauma is the most common indication for iris-claw IOL in pediatrics, followed by ectopia lentis and by congenital cataract. Iris-claw IOL was implanted anteriorly in 70.7% of pediatrics. Patients with retropupillary position achieved better visual outcome. Anterior iris-claw IOL had higher intraocular pressure readings and more decentration and disenclavation. In comparison with the adult group consisting of 130 eyes, pediatric iris-claw implantation was associated with more decentration and disenclavation, while adult practice was associated with more macular edema. **Conclusions:** Iris-claw IOL offers a relatively safe method for optical rehabilitation in pediatric patients where the capsular support is inadequate. Retropupillary implantation may achieve better visual outcome with less frequency of glaucoma and corneal endothelial loss.

## 1. Introduction

Dealing with childhood aphakia and its subsequent complications and optical rehabilitation, in the presence of an unstable capsule-zonular complex, is one of the most challenging aims encountered by the ophthalmologist [1,2,3,4]. The choice and technique of intraocular lenses when operating on pediatric eyes is still controversial. Proper high refractive error correction of the aphakia is the primary goal to diminish amblyopia. The most useful method to achieve this goal is by implantation of an intraocular lens (IOL), which averts the children and their parents from using contact lenses or the wearing of heavy, cosmetically unacceptable spectacles [1,5]. With increasing experience, several complications and problems in implant surgery in children have now become apparent. The pediatric eye shows more inflammatory reaction after surgery compared to an adult eye and tends to behave differently to surgical intervention. Special biochemical and anatomical aspects of the juvenile eye require certain technical adaptations during surgery [5,6].

In cases of congenital cataract extraction with intact capsular bag, in-the-bag implantation of a posterior chamber IOL is preferable and first line in almost all cases [7,8,9]. However, the capsular bag support could be deficient or risky to implant the IOL in the bag or the sulcus in many situations, such as traumatic or spontaneous crystalline lens subluxation, complicated cataract surgery, IOL dislocation, zonular dehiscence, and postoperative aphakic secondary implantation after early congenital cataract extraction [10,11,12,13,14]. In order to overcome this problem, several techniques were utilized such as angle-supported anterior chamber implantation, IOL scleral fixation, and iris-claw IOL implantation, either in the anterior chamber or in the retropupillary space [10,11,12,13,14].

Turnen et al. [15] reported a high reoperation rate of 30% with anterior chamber IOLs. Additionally, higher risks of damage to the corneal endothelium and trabecular meshwork, with subsequent endothelial cell loss, corneal decompensation, and glaucoma have been reported [16]. Anterior chamber IOLs can be related to another complications such iris sphincter erosion, chronic inflammation, and hyphema, the triad called Uveitis–Glaucoma–Hyphema Syndrome [17]. Scleral-fixated IOLs have a high rate of complications, including breakage, suture erosion, and lens subluxation, as well as suprachoroidal hemorrhage, with early IOL dislocation reported at 29%. In addition, these IOLs are associated with hypotony and collapse during the passage of scleral sutures. Moreover, anterior chamber collapse can lead to corneal endothelial loss as well as compromise the exact point of needle [1]. Sutureless intrascleral fixated IOLs are widely adopted for adults. However, the implantation of these types of IOL is challenging because of the elastic sclera and smaller eyes in pediatric patients. In all types of implantations, IOL tilt, decentration, and subluxation are also major complications with the pediatric eyes [18,19].

The first model of an iris-claw IOL was used in 1972 with fixation the IOL to the midperipheral, less vascularized area of the iris [11,20]. The most popular version of iris-claw IOLs is the Artisan Aphakia Model (convex/concave) (Ophtec BV, Groningen, the Netherlands). Although this Artisan IOL model was designed to be implanted in the anterior chamber and showed favorable visual outcome with low rate of complications [11,21,22], many surgeons investigated the implantation of this IOL in the retropupillary space due to the possible risk of endothelial cell loss associated with anterior chamber implantation [11,13,19].

The aims of this study were to evaluate the overall practice of iris-claw IOL implantation in children, to compare the visual outcomes and postoperative complication rates between children and adults, and assess the differences between anterior chamber and retropupillary implantation.

## 2. Methods

### 2.1. Patients and Data

After obtaining the Institutional Review Board (IRB) approval, we retrospectively reviewed the demographic and clinical characteristics of 41 eyes of 34 children and 130 eyes of 117 adults who underwent iris-claw “Artisan^®^” IOL implantation during the period between January 2014 to July 2020. The study was conducted at a tertiary ophthalmic center located in the north of Jordan. Using the medical records, demographic data (age, sex), past ocular history, and past medical history were collected. The indications for iris-claw IOL implantation, preoperative optical parameters, intraoperative details, visual outcome, and postoperative complications were also retrieved.

All cases of primary or secondary iris-claw IOL implantation regardless of the indication were included in the study. Patients with insufficient preoperative or postoperative data were excluded. Patients were categorized by age group (pediatric: <18 years, adults: ≥18 years), location of implantation (anterior iris-claw IOL or retropupillary iris-claw IOL), and the indication for surgery.

Data on postoperative complications comprised iris tissue loss, irregular iris shape, spontaneous disenclavation of one of the haptics or both, iris-claw IOL decentration, endothelial cell loss (only clinically assessed by the clinical appearance of persistent corneal edema and the development of bullous keratopathy), postoperative high intraocular pressure (IOP), retinal detachment, macular edema, endophthalmitis, and epiretinal membrane proliferation.

### 2.2. Indications for Surgery

The indications for surgery were categorized into 5 groups. Group 1 included traumatic cases with either ruptured cataractous inadequately-supported lens or lens subluxation. Group 2 comprised cases of complicated cataract surgery resulting in posterior capsule rupture (PCR) or zonular dialysis with inadequate capsular support, dropped IOL, and subluxated IOL. The iris-claw IOL was implanted either as a primary IOL implantation intraoperatively during complicated cataract surgery or as a secondary IOL implantation in another surgery. Group 3 was the pediatric cases of congenital cataract extraction without adequate capsular support. Group 4 included cases with non-traumatic ectopia lentis, mainly as a result of connective tissue diseases such Marfan’s syndrome. Finally, group 5 included cases which previously underwent anterior chamber (AC) IOL with angle support which subsequently developed ocular complications such as secondary glaucoma or bullous keratopathy.

### 2.3. Ocular Parameters

The optical parameters included the iris-claw IOL power (using an A-constant of 115.0 for anterior iris-claw IOL and 116.9 for retropupillary iris-claw IOL), keratometry readings (using keratometry), and axial length (through applanation ultrasound measurement). The visits at 1 week, 1 month, 3 months, 1 year, and at the last follow postoperatively were assigned for data assessment.

The outcome was compared between both main groups (retropupillary versus anterior iris-claw, and pediatrics versus adults) for visual outcomes and postoperative complications. The visual acuity was measured in decimal visual acuity (through E chart) and converted to LogMAR visual acuity. The best correct visual acuity (BCVA) was used in the analysis. For patients with visual acuity of counting fingers, hand motion, light perception or “no light perception”, they were converted according to Schulze-Bonsel K et al. [23].

### 2.4. Surgical Technique and Postoperative Care

The choice of whether to implant the iris-claw IOL anteriorly or retropupillary is individualized and depends on the specific ocular condition and the experience of seven consultant surgeons who performed the surgeries. They follow the same standardized institutional procedural guidelines with the same instrumental machines. The type of iris-claw IOL utilized in this study was the Artisan^®^ aphakia IOL (Ophtec BV, Groningen, The Netherlands). The SRK/T formula was the most common formula utilized to determine the desired IOL power. Haigis formula was used in high myopia, and Holladay II and Hoffer Q were utilized for short axial lengths with the aim of achieving slight postoperative myopia except in certain pediatric patients. For both groups (retropupillary and anterior iris-claw IOLs), the operations were performed under either general or local anesthesia. The operations started with the creation of two corneal side ports. After performing the primary associated procedure, miosis was induced through the intracameral injection of acetylcholine 1%. A 5.5-mm clear corneal incision was made at 12 o’clock position. It was constructed using 2.8 keratome through the para-limbal space and extended to up 5.5 mm.

For retropupillary “posterior” implantation, the iris-claw IOL was rotated upside down (with its convex surface toward posteriorly). The process of rotation was carried out by a special forceps. After that, a long micro-spatula was then utilized through the side port to tuck iris tissue into the claws.

Regarding the anterior implantation, the convex surface faced anteriorly, and the iris was tucked at the iris midperiphery between the claw haptics. The corneal incision was closed using three interrupted 10-0 nylon sutures. At the end of the surgery, prophylactic 0.1 mL intracameral moxifloxacin was administered.

The postoperative course included steroid, antibiotics, and nonsteroidal anti-inflammatory eye drops. For pediatrics, oral prednisolone was prescribed in some cases due to the intense inflammatory response.

### 2.5. Statistical Analysis

The collected data was entered into a spreadsheet and analyzed using the IBM SPSS statistical package for Windows v.26 (Armonk, NY, USA). Nominal variables were expressed as the frequency (percentage) and continuous variables as the mean ± standard error of the mean (SEM). Data normality was tested using the Kolmogorov–Smirnov test. All variables were normally distributed. The statistical significance between the study groups was determined by using the Chi-square test for categorical variables and the Student’s *t*-test for continuous variables. A statistically significant result was considered if *p* ≤ 0.05. The sample size was confirmed retrospectively at alpha level of 0.05 and power of analysis at 90%.

## 3. Results

### 3.1. General Characteristics of Pediatric Patients

In this study, 41 eyes of 34 pediatric patients were included. Of those, 56.1% were males. The mean age of the patients was 11.1 years. The left eye was operated on in 22 (53.7%) procedures. Iris-claw IOL was implanted anteriorly in 29 (70.7%) of the eyes. The mean follow-up period was 24.5 months.

The most common indication for iris-claw IOL in pediatric patients was trauma which was performed in 12 (29.3%) eyes followed by ectopia lentis in 11 (26.8%) eyes, congenital cataract in 9 (22.0%) eyes, AC IOL related complications with subsequent exchange in 5 patients, and following complicated cataract surgery in 4 eyes.

Pre-existing retinal detachment and preoperative glaucoma were found in three and eleven patients, respectively. The iris-claw IOL implantation was done along with PI in 63.4% of the eyes. Decentration of the iris-claw IOL and disenclavation of one of the haptics were the two most common complications after iris-claw IOL implantation in pediatric patients. Postoperative high IOP developed in 9 (22.0%) patients, and signs of corneal endothelial damage were observed in 8 (19.5%) of the patients. Two cases of retinal detachment were reported. No cases of macular edema or endophthalmitis were reported. The mean change in the BCVA at one month was −0.2400 LogMAR (which is equivalent to an improvement in 12 letters). Table 1 summarizes the overall characteristics of the study cohort.

### 3.2. Retropupillary Versus Anterior Artisan in Pediatric Patients

There was no difference between retropupillary and anterior iris-claw IOL in terms of sex, age, laterality, and comorbidities. Regarding the indications, anterior iris-claw IOL was performed significantly more in trauma patients, while the retropupillary iris-claw IOL was done more in patients following AC IOL-related complications. Preoperative glaucoma was found to be more common in the retropupillary group. Regarding the associated surgical procedure, PI was performed more when the implantation was anterior. In addition, IOL exchange was achieved more in the retropupillary location.

The visual outcome was statistically different between both groups, and Table 2 shows the difference in the visual outcome. The mean change in BCVA at 1-month postoperative period was higher in the retropupillary group (mean change of −0.7000 LogMAR, improvement of 35 letter) than the anterior group (mean change of −0.0429, improvement of 21 letter). In addition, the mean change in the BCVA at 1 year postoperatively was significantly higher in the retropupillary group, which achieved better improvement (mean change of BCVA of −0.6143 LogMAR, improvement of 31 letter) than anterior group (mean change of BCVA of −0.2190 LogMAR, improvement of 11 letters). A similar pattern was found in the mean change of BCVA at the last follow-up visit.

Multiple linear regression analysis was performed to assess the factors that affected visual acuity independently (location, age, gender, previous ocular diseases and surgery, and indications). The location of the iris-claw IOL was the only independent factor affecting the visual acuity outcome (being a retropupillary implant was associated with better outcome) (B value = 0.984, confidential interval for B: 0.064–1.904, *p* value = 0.039).

The postoperative complications varied between both groups as most complications developed when the iris-claw IOL was implanted anteriorly. Both iris-claw IOL decentration and disenclavation were more significantly associated with anterior Artisan implantation. Furthermore, high IOP and/or the prolonged use of postoperative antiglaucoma agents were associated with anterior implantation (although this was not statistically significant). Similarly, signs of corneal endothelial loss were higher in the anterior location. Table 2 summarizes the differences between anterior and retropupillary iris-claw IOL.

### 3.3. Pediatric Versus Adult Iris-Claw Practice

There was no difference between the adult and pediatric groups in terms of gender, laterality, or the location of the iris-claw IOL (anterior or retropupillary). Regarding the indications, trauma, congenital cataract, and ectopia lentis were more significantly associated with the pediatric group. On the other hand, complicated cataract surgery was more common in the adult group. The associated ocular conditions varied between both groups. As expected, pseudoexfoliation syndrome and diabetic retinopathy were reported only in the adult group. Regarding the associated ocular procedure, PPV and IOL exchange were performed more in the adult group.

The postoperative complications varied between both groups. IOL decentration and disenclavation were observed more in the pediatric group. There was no difference between adult and pediatric groups in terms of high postoperative IOP, retinal detachment, and corneal endothelial loss. On the other hand, macular edema was reported exclusively in the adult group. The mean change in the BCVA was not significantly different between both groups. Table 1 summarizes the differences between adult and pediatric groups.

## 4. Discussion

Previous studies have shown the safety and efficacy of iris-claw lens implants in adults [11,13,19,24,25,26]. However, most studies on pediatric eyes with long-term results were limited to single case reports, included low numbers of patients, or were limited to only one indication [6,7,27,28,29,30]. In the current study, we included all cases of iris-claw implantation for all possible indications in pediatric patients and investigated the effect of all associated procedures, ocular diseases, and the location of the IOL. Furthermore, we compared the study cohort to an adult cohort from a previously published study by the authors at the same hospital setting. We also included a relatively good sample size. We found that pediatric patients underwent anterior implantation of iris-claw IOL more commonly for traumatic indications, while the retropupillary implantation was used more commonly after cases with AC IOL complications. Moreover, retropupillary implantation was the only independent factor affecting visual outcomes. Patients with anterior implantation of iris-claw IOL developed more disenclavation; however, it was safer for patients with ectopia lentis. Macular edema developed exclusively in adult patients.

One concern about iris-claw IOLs is the possible long-term influence on the corneal endothelium. Children have a long mean life expectancy, and any cataract operation may adversely affect endothelial cell [10]. Güell et al. reported that adult anterior iris-claw IOL implantation caused 10.9% endothelial cell loss at 3 years follow-up [10], while Anbari and Lake reported a mean drop of 267 cells/mm^2^ (about 11.7%) of endothelial cell density at 2 years after retropupillary iris-claw IOL implant [31]. In pediatric patients, Fuerst et al. reported an average of 1.34% annual decrease in corneal endothelial cell count, with a mean follow-up of 3.59 years after anterior implantation of iris-claw IOL for ectopia lentis [27]. Lifshitz et al. found no significant difference in corneal endothelial count between the operated eye and the unoperated fellow eye in two children after lens removal and anterior implantation of iris-claw IOL for subluxated crystalline lenses after eight months of follow-up [6]. This was also reported by Sminia et al. in their study regarding anterior iris-claw IOL implantation in pediatric patients with cataract, where they found no significant drop of the endothelial cells count [7]. Yueqin C et al. proposed that corneal endothelial cell loss may be due to a mechanical irritation between the endothelium and the instruments or the IOL haptics; therefore, using sufficient amount of viscoelastic material during surgery may minimize endothelial cells loss [32]. In this study, about one fifth of pediatric patients developed signs of endothelial cells loss. However, except for one case, all cases of pediatric endothelial cell loss developed in anteriorly implanted iris-claw IOL. There was no difference between pediatrics and adults in terms of endothelial cell loss.

Ectopia lentis is a well-recognized ocular manifestation of Marfan syndrome; it may also occur in association with other rare systemic diseases or in a hereditary fashion. Placement of the IOL in the capsular bag is not an option available to many patients with ectopia lentis because of inadequate zonular support [12]. Iris-claw IOL may therefore be an optimal option for Marfan’s syndrome patients with minimal complications regardless of its location [12]. In their randomized trial, Hirashima et al. studied 31 eyes of 16 patients with ectopia lentis due Marfan’s syndrome and found that the improvement in visual acuity was similar in both retropupillary and anterior chamber implantation; however, IOL disenclavation tended to occur more frequently in the retropupillary group, though the difference was not significant [33]. We recently demonstrated a similar improvement in visual acuity with both retropupillary and anterior chamber implantation in adult patients; however, IOL disenclavation was seen more in the anterior group [12]. In this study, the results demonstrated that the rate of iris-claw IOL disenclavation was significantly higher in pediatrics than adults. This may be attributed to the inclusion of Marfan’s syndrome patients within the pediatrics group. Furthermore, pediatric patients have growing eyes, which may predispose them to higher rates of disenclavation.

Macular edema is a known complication after iris-claw implantation [19,34]. The exact mechanism is not well understood but may be the result of chronic irritation of the iris or resul from the primary cause of aphakia [35]. The risk of macular edema appeared to be higher with anterior IOL implantation in other studies [19]. In the current study, macular edema developed more after anterior implantation in adults than retropupillary implantation, possibly due to different severity of induced postoperative inflammation and pigment dispersion between the anterior and posterior iris surfaces. Another explanation is the iris-claw IOL movement with subsequent iris irritation in the anterior iris-claw IOL group resulting from insufficient iris tissue and insufficient capturing of the claw compared to the retropupillary location where the IOL might be able to enclave more iris tissue [36]. Regarding the exclusive development of macular edema in adults, the low prevalence of macular edema in children may be attributed to a number of factors, including superior structural integrity of blood vessel walls compared to that of adult vasculature, differences in prostaglandins, and operative techniques [37,38]. Moreover, it is well known that vitreous loss and traction resulted from capsule rupture during complicated cataract surgery is now a well-recognized risk factor for the development of macular edema [37,39]. The complicated cataract surgery (with more chance of vitreous traction) and subsequent iris-claw IOL implantation was performed more in the adult group in this study.

Regardless of the site of implantation (anteriorly or retropupillary), iris-claw IOLs have several drawbacks. The 5.5 mm large corneal incision with the consequent corneal astigmatism is a major limitation. Baykara et al. preferred a scleral tunnel incision that normally does not require sutures and reduces postoperative astigmatism [40]. For the retropupillary implantation, IOL dislocation into the vitreous due to enclavation failure could be a major complication. However, these cases can often be treated by re-enclavation [19].

Many limitations were elaborated from this study. First, our inability to assess endothelial cell count objectively decreases the ability to measure the influence of the location of implantation on corneal endothelium. The study lacks long-term ocular parameter and safety data on iris-claw IOL. Second, the retrospective nature of the study along with the unequal size of both groups is another limitation and increases the complexity of the analysis. Efficacy and safety of iris-claw IOL in pediatrics would not be concluded from retrospective studies. Third, diverse indications included in the analysis enhanced the novelty of the paper; however, at the same time, this would raise the heterogenicity of the analysis. Finally, multiple surgeons could be a source of another heterogenicity.

## 5. Conclusions

Iris-claw IOL was be considered as a solution for pediatric patients without adequate capsular support to achieve better visual outcome. This study revealed that in pediatric patients, the retropupillary iris-claw IOL may be suitable for most patients with a wide range of indications and with satisfactory clinical outcome. Further prospective studies and randomized trials on larger samples are needed to investigate the best approach in patients with inadequate capsular support and to establish the results of these retrospective studies.

## Figures and Tables

**Table 1 jcm-14-01135-t001:** General demographic and clinical characteristics of pediatric and adult patients.

Variable	Pediatric Patients (%) (N = 41)	Adult Patients (%) (N = 130)	*p*-Value
Sex			
Male	23 (56.1)	79 (60.8)	NS
Female	18 (43.9)	51 (39.2)	
Age (years) *	11.1 ± 0.8	55.8 ± 1.8	0.0001
Location of implantation			
Posterior “Retropupillary”	12 (29.3)	49 (37.7)	NS
Anterior	29 (70.7)	81 (62.3)	
Side of procedure (laterality)			
Right (OD)	19 (46.3)	63 (48.5)	NS
Left (OS)	22 (53.7)	67 (51.5)	
Indications			
Trauma	12 (29.3)	10 (7.7)	0.0001
Complicated surgery	4 (9.7)	94 (72.4) ↑	
Congenital cataract	9 (22.0) ↑	5 (3.8)	
Ectopia lentis	11 (26.8) ↑	12 (9.2)	
AC IOL complications	5 (12.2)	9 (6.9)	
Associated ocular diseases			
Diabetic retinopathy	0 (0.0)	21 (16.2)	0.0001
Retinal detachment	3 (7.3)	14 (10.9)	NS
Preoperative glaucoma	11 (26.8)	21 (16.2)	NS
Previous surgical ocular history:			
Primary traumatic repair	11 (26.8)	22 (16.9)	NS
Keratoplasty	2 (4.9)	4 (3.1)	NS
Pars plana vitrectomy	5 (12.2)	26 (20.0)	NS
Phacoemulsification	12 (29.3)	55 (42.3)	NS
Congenital cataract extraction	9 (22.0)	6 (4.6)	0.005
Associated ocular procedure (during artisan implantation):			
Peripheral iridectomy	26 (63.4)	80 (61.5)	NS
IOL exchange	7 (17.1)	44 (33.8)	0.029
Pars plana vitrectomy	2 (4.9)	31 (23.8)	0.004
Keratoplasty	2 (4.9)	5 (3.8)	NS
Phacoemulsification	14 (34.1)	28 (21.5)	NS
Congenital cataract extraction	3 (7.3)	3 (2.3)	NS
Anterior vitrectomy	20 (48.8)	62 (47.7)	NS
Silicon oil removal	2 (4.9)	6 (4.6)	NS
Pupilloplasty	1 (2.4)	4 (3.1)	NS
Synechiolysis	1 (2.4)	7 (5.4)	NS
Postoperative complications			
Irregular pupil shape	4 (9.8)	27 (21.3)	0.073
Iris atrophy	5 (12.2)	23 (18.1)	NS
Decentration or tilt	13 (31.7)	13 (10.2)	0.002
Disenclavation	11 (26.8)	6 (4.7)	0.003
Signs of corneal endothelial loss	8 (19.5)	17 (13.4)	NS
Pigment dispersion	4 (9.8)	8 (6.3)	NS
Postoperative high IOP and/or use of new antiglaucoma and/or glaucoma surgery	9 (22.0)	29 (22.8)	NS
Hypotony	1 (2.4)	7 (5.5)	NS
Retinal detachment	2 (4.9)	5 (3.9)	NS
Epiretinal membrane proliferation	0 (0.0)	3 (2.4)	NS
Macular edema	0 (0.0)	15 (11.8)	0.012
Endophthalmitis/keratitis	0 (0.0)	2 (1.6)	NS
Artisan calculation characteristics *			
Axial length (millimeter)	23.86 ± 0.4	23.66 ± 0.2	NS
Km (diopter)	42.44 ± 0.5	45.38 ± 0.3	0.012
IOL power (diopter)	18.2 ± 0.3	18.0 ± 0.3	NS
Follow up period (months) *	24.5 ± 2.1	14.9 ± 1.4	0.02
Overall visual outcome *			
Change in BCVA at 1 week postoperative (LogMAR)	0.0652 ± 0.08	−0.0793 ± 0.07	NS
Change in BCVA at 1 month postoperative (LogMAR)	−0.2400 ± 0.07	−0.3707 ± 0.07	NS
Change in BCVA at 3 months postoperative (LogMAR)	−0.1552 ± 0.09	−0.42 0.147 ± 0.08	NS
Change in BCVA at 1 year postoperative (LogMAR)	−0.3179 ± 0.1	−0.3305 ± 0.1	NS
Change in BCVA at last follow up (LogMAR)	−0.1925 ± 0.06	−0.3359 ± 0.1	NS

Abbreviations—OD: right eye; OS: left eye; IOL: intraocular lens; BCVA: best corrected visual acuity; Km: average keratometry readings; AC: anterior chamber; IOP: intraocular pressure; NS: not significant; ↑: significant increase *: mean ± standard error of the mean.

**Table 2 jcm-14-01135-t002:** Posterior versus anterior iris-claw IOL.

Variables	Posterior “Retropupillary” Artisan (%)	Anterior Artisan (%)	*p*-Value
Sex			NS
Male	5 (41.7)	18 (62.1)	
Female	7 (58.3)	11 (37.9)	
Age (years) *	11.8 ± 1.6	10.8 ± 0.9	NS
Side of procedure (laterality)			NS
Right (OD)	6 (50.0)	13 (44.8)	
Left (OS)	6 (50.0)	16 (55.2)	
Indications			0.049
Trauma	1 (8.3)	11 (37.9) ↑	
Complicated surgery	2 (16.7)	2 (6.9)	
Congenital cataract	4 (33.3)	5 (17.2)	
Ectopia lentis	2 (16.7)	9 (31.0)	
AC IOL complications	3 (25.0) ↑	2 (6.9)	
Associated ocular diseases			
Diabetic retinopathy	0 (0)	0 (0)	NS
Retinal detachment	0 (0)	3 (10.3)	NS
Preoperative glaucoma	7 (58.3)	4 (13.8)	0.007
Previous surgical ocular history:			
Primary traumatic repair	1 (8.3)	10 (34.5)	NS
Keratoplasty	1 (8.3)	1 (3.4)	NS
Pars plana vitrectomy	0 (0.0)	5 (17.2)	NS
Phacoemulsification	5 (41.7)	7 (24.1)	NS
Congenital cataract extraction	5 (41.7)	7 (24.1)	NS
Anterior vitrectomy	4 (33.3)	7 (24.1)	NS
Associated ocular procedure (during artisan implantation:			
Peripheral iridotomy	3 (25.0)	22 (81.6)	0.001
IOL exchange	5 (41.7)	2 (6.9)	0.016
Pars plana vitrectomy	1 (8.3)	1 (3.4)	NS
Keratoplasty	1 (8.3)	1 (3.4)	NS
Phacoemulsification	3 (25.0)	11 (37.9)	NS
Congenital cataract extraction	2 (16.7)	1 (3.4)	NS
Anterior vitrectomy	6 (50.0)	14 (48.3)	NS
Pupilloplasty	1 (8.3)	0 (0)	NS
Synechiolysis	0 (4.9)	1 (3.4)	NS
Postoperative complications			
Irregular iris shape	0 (0.0)	4 (13.8)	NS
Iris atrophy	1 (8.3)	14 (13.8)	NS
Artisan decentration or tilt	0 (0.0)	13 (44.8)	0.004
Disenclavation	1 (8.3)	10 (34.5)	0.034
Signs of corneal endothelium loss	1 (8.3)	7 (24.1)	NS
Pigment dispersion	0 (0)	4 (13.8)	NS
Postoperative high IOP and/or use of newantiglaucoma and/or glaucoma surgery	2 (16.7)	7 (24.1)	NS
Hypotony	0 (0.0)	1 (3.4)	NS
Retinal detachment	0 (0.0)	2 (6.9)	NS
Follow up period (months) *	19.5 ± 5.1	26.4 ± 3.6	NS
Overall visual outcome (Mean in LogMAR) *			
Change in BCVA at 1 week postoperative (LogMAR)	−0.1200 ± 0.1	0.2077 ± 0.08	NS
Change in BCVA at 1 month postoperative (LogMAR)	−0.7000 ± 0.1	−0.0429 ± 0.08	0.022
Change in BCVA at 3 months postoperative (LogMAR)	−0.3875 ± 0.1	−0.0667 ± 0.07	NS
Change in BCVA at 1 year postoperative (LogMAR)	−0.6143 ± 0.1	−0.2190 ± 0.09	0.019
Change in BCVA at last follow up (LogMAR)	−0.4083 ± 0.1	−0.1000 ± 0.07	0.034

Abbreviations:—OD: right eye; OS: left eye; IOL: intraocular lens; BCVA: best corrected visual acuity; AC: anterior chamber; IOP: intraocular pressure; NS: not significant. ↑: significant increase *: mean ± standard error of the mean.

## Data Availability

Data is available upon request to correspondence author.

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
