# Peer review of "Iris-Claw Intraocular Lens Implantation in Pediatric Population: Indications, Outcomes, and a Comparison with Adult Population"

_jcm, 2025, doi:10.3390/jcm14041135_

Round 1
Reviewer 1 Report
Comments and Suggestions for Authors
Dear authors, Thanks for the opportunity to review this interesting paper.
I have included comments & questions in the pdf version of your paper.
Kind regards.

Author Response
Editor-in-Chief
Journal of Clinical Medicine
Re: Iris-claw intraocular lens implantation in pediatric patients: indications and visual outcomes. Manuscript ID: jcm-3472542
Dear Editor and Reviewers,
My self and the co-authors are pleased to resubmit the manuscript entitled ‘Iris-claw intraocular lens implantation in pediatric patients: indications and visual outcomes” to be considered for publication in Journal of Clinical Medicine. The revised manuscript takes into consideration both the editorial and reviewers’ comments. Kindly find below a point-by-point response to those comments, along with an uploaded copy marked with MS track changes indicating changes to the manuscript.
Response to reviewer 1
Many thanks for your valuable comments and remarks, the authors will address each comment separately, and alterations will be amended in the revised manuscript accordingly:
- " Regarding Follow-up time in the results of the abstract" Thank you very much for your comprehensive point. We have added the follow up time for the abstract.
- “Lacks endothelial cell follow-up data.” Thank you very much again for this important point. Unfortunately, this is one of the important limitations to our study. In our institution, there is no specular microscopy that would enable us to assess the endothelial cell count. We have assessed the endothelial function clinically. This point is added to the limitations of the revised manuscript.
- “ High heterogenicity, but clinical experience regarding the indications for surgery.” Thank you very much for this meaningful point. I would seem that the heterogenicity for the indications is not useful. However, we aimed to investigate the practice of iris-claw IOL among all possible indications. To solve this heterogeneity, we have performed multivariate logistic regression analysis to justify the independent factors affecting the visual outcome and reduce the effect of the heterogeneity. However, we have elaborated on this point also in the limitations. Thank you so much.
- “Revisar K’ Thank you so much. We have revised the A constant into 116.9.
- “High heterogenicity regarding seven consultant surgeon” Thank you so much. We agree with that. The heterogenicity was reduced as all of them follow the same institutional and procedural guidelines with the same instrumental machines. However, we have added this comment into the limitations.
- “Scleral incision?” Thank you so much for your kind comment. The incision for implantation the iris-claw IOL was clear corneal incision about 5.5 mm. It was constructed using 2.8 keratome through the para-limbal space and extended to up 5.5 mm.
- “Conclusion without evidence nor scientific value” Thank you so much. We have modified the conclusions. Thank you so much.

Reviewer 2 Report
Comments and Suggestions for Authors
This retrospective study evaluates the use of iris-claw intraocular lenses (IOLs) in pediatric patients with capsular insufficiency, comparing visual outcomes and complications with adults. The study included 41 eyes from 34 pediatric patients who underwent Artisan® iris-claw IOL implantation between January 2014 and July 2020. The most common indications were trauma, ectopia lentis, and congenital cataract. Anterior implantation was used in 70.7%, while retropupillary positioning provided better visual outcomes and fewer complications. Pediatric patients experienced more lens decentration and disenclavation, while adult patients faced more macular edema. The study concludes that iris-claw IOLs are a safe option for optical rehabilitation in pediatric patients, with retropupillary implantation offering superior outcomes, reduced glaucoma risk, and less corneal endothelial damage. While pediatric cases show more instability, adults may encounter different complications like macular edema. The study was compliant with an approved IRB protocol and requirement for signed informed consent was waived due to retrospective nature of the study.
Major:
1. The study retrospectively enrolled pediatric and adult patients of both sexes and enrolled adults cases appear almost 10X higher that the pediatric cases (Table 1). Why the title of the manuscript did remain silent on mentioning adult information?
2. How were the p values calculated for sex distribution? Was it a side-by-side comparison between same sexes in both pediatric and adult groups or was that the overall cases of both sexes in pediatric and adult group (Table 1 and table 2)? The same concern applies to OD vs OS data in Table 1 and Table 2.
3. The retrospective study design does not seem to be the best way of testing the safety and efficacy and visual outcomes of the Iris-claw IOL implantation. A prospectively enrolled, randomized and single‑blinded study design would have been more appropriate. Mixing pediatric and adult cases for the enrollment purpose makes the study design and interpretation very complicated. The biased study design has also been identified as a major drawback.
4. The study only talks about surgical technique and the postoperative care. The study is seriously limited with post-operative systematic review of the follow up data over a prolonged period to make any firm conclusion supporting the aim of the study. The lack of long-term ocular parameter and safety data on iris-claw IOLs in children warrants caution, particularly concerning potential endothelial cell loss, which can lead to corneal decompensation. Therefore, further high-quality, interventional, long-term research is needed to better understand the safety, efficacy, and potential risks of iris-claw IOLs in children, ensuring their optimal use and minimizing any potential adverse effects.
Minor:
1. Table 1, Page 5: Correct ‘trauma’ to ‘Trauma’
Author Response
Editor-in-Chief
Journal of Clinical Medicine
Re: Iris-claw intraocular lens implantation in pediatric patients: indications and visual outcomes. Manuscript ID: jcm-3472542
Dear Editor and Reviewers,
My self and the co-authors are pleased to resubmit the manuscript entitled ‘Iris-claw intraocular lens implantation in pediatric patients: indications and visual outcomes” to be considered for publication in Journal of Clinical Medicine. The revised manuscript takes into consideration both the editorial and reviewers’ comments. Kindly find below a point-by-point response to those comments, along with an uploaded copy marked with MS track changes indicating changes to the manuscript.
Response to reviewer 2
Many thanks for your valuable comments and remarks, the authors will address each comment separately, and alterations will be amended in the revised manuscript accordingly, again, thank you so much for the efforts in improving the manuscript
- " The study retrospectively enrolled pediatric and adult patients of both sexes and enrolled adults cases appear almost 10X higher that the pediatric cases (Table 1). Why the title of the manuscript did remain silent on mentioning adult information? " Thank you very much for your comprehensive comment. We have modified the title of the revised manuscript according to this comment.
- “How were the p values calculated for sex distribution? Was it a side-by-side comparison between same sexes in both pediatric and adult groups or was that the overall cases of both sexes in pediatric and adult group (Table 1 and table 2)? The same concern applies to OD vs OS data in Table 1 and Table 2.” Thank you very much again for this important question. It is a side-by-side comparison between the same sex (or the same laterality) among both pediatrics and adults. Accordingly, for example, the pediatric male’s number (56% from the whole pediatrics 23/41) is similar to the adult males (60.8% from the whole adults 79/130). So, the p value was not significant. A chi-square test was used. The test compares the proportions of males and females within each group (pediatric and adult).
- “ The retrospective study design does not seem to be the best way of testing the safety and efficacy and visual outcomes of the Iris-claw IOL implantation. A prospectively enrolled, randomized and single‑blinded study design would have been more appropriate. Mixing pediatric and adult cases for the enrollment purpose makes the study design and interpretation very complicated. The biased study design has also been identified as a major drawback.” Thank you so much for your valuable point. We agree with completely. Prospective studies and the trials are more appropriate to build up strong conclusions. Retrospective studies are the basis to make a theory regarding your practice. Now, we are working on another prospective randomized trial for the efficacy, safety and outcomes of iris-claw IOL but it needs lengthy time. We conducted this study to visualize our experience. We have modified the conclusions of this study to reduce the tone of the conclusions. Moreover, we have added this point to the limitations.
- ” The study only talks about surgical technique and the postoperative care. The study is seriously limited with post-operative systematic review of the follow up data over a prolonged period to make any firm conclusion supporting the aim of the study. The lack of long-term ocular parameter and safety data on iris-claw IOLs in children warrants caution, particularly concerning potential endothelial cell loss, which can lead to corneal decompensation. Therefore, further high-quality, interventional, long-term research is needed to better understand the safety, efficacy, and potential risks of iris-claw IOLs in children, ensuring their optimal use and minimizing any potential adverse effects.” Thank you again. Thank you for this meaningful point. We also agree with that. This is the main limitation of any retrospective study. We are working on further prospective trials. Furthermore, we have modified the conclusions of the manuscript.
- “Table 1, Page 5: Correct ‘trauma’ to ‘Trauma’” Thank you. It was modified.
Again and again, thank you from the heart for your efforts in improving the manuscript.

Reviewer 3 Report
Comments and Suggestions for Authors
Dear Authors,
This is interesting manuscript concern important problem of lens implantation in pediatric population. I have only a few remarks :
1. Introduction – please add more about used lens, advantages and disadventages of different types of lens for pediatric population, and possible problems.
2. Statistical analysis- You use ANOVA and t-student’s test- what needs normality of distribution of all of the parameters – please add, what test was used to check the normality, and sentence that all parameters have normality in distribution and uniformity of variancy. If not change the statistic test.
3. In introduction You wrote that in children population more infectious complications were observed, but not in Your study- did You used another type of prophylaxis- please explain in manuscript.
4. Discussion – discuss more with Your results, not only describe other studies.
5. Conclusions line 323- What do You mean?
6. Author contribution- it should be filled in.
Author Response
Editor-in-Chief
Journal of Clinical Medicine
Re: Iris-claw intraocular lens implantation in pediatric patients: indications and visual outcomes. Manuscript ID: jcm-3472542
Dear Editor and Reviewers,
My self and the co-authors are pleased to resubmit the manuscript entitled ‘Iris-claw intraocular lens implantation in pediatric patients: indications and visual outcomes” to be considered for publication in Journal of Clinical Medicine. The revised manuscript takes into consideration both the editorial and reviewers’ comments. Kindly find below a point-by-point response to those comments, along with an uploaded copy marked with MS track changes indicating changes to the manuscript.
Response to reviewer 3
Many thanks for your valuable comments and remarks, your comments were crucial in improving our manuscript. The authors will address each comment separately, and alterations will be amended in the revised manuscript accordingly:
- " Introduction – please add more about used lens, advantages and disadvantages of different types of lens for pediatric population, and possible problems." Thank you very much. We have added the advantages and disadvantages of different types of IOL into the revised manuscript.
- “Statistical analysis- You use ANOVA and t-student’s test- what needs normality of distribution of all of the parameters – please add, what test was used to check the normality, and sentence that all parameters have normality in distribution and uniformity of variancy. If not change the statistic test.” Thank you very much again for this important and crucial point. The normality of the data was tested using the Kolmogorov-Smirnov test. All continuous variables were normally distributed. We have the required modifications.
- In introduction You wrote that in children population more infectious complications were observed, but not in Your study- did You used another type of prophylaxis- please explain in manuscript.” Thank you very much. The reported sentence in the Introduction is “The pediatric eye shows more inflammatory reaction after surgery compared to an adult eye”. We reflected the intense inflammatory uveitis reaction (not the infectious) in pediatrics. However, we have added to the revised manuscript the infectious prophylaxis and the medications utilized for the uveitis reactions also.
- “Discussion – discuss more with Your results, not only describe other studies..” Thank you very much. We have elaborated more about our results in the revised manuscript.
- Conclusions line 323- What do You mean?.” Thank you very much. It is an error and correct to “Iris-claw IOL is a proper solution for pediatric patients without adequate capsular support to achieve better visual outcome”
- “Author contribution- it should be filled in..” Thank you so much. We have filled the author contributions section.
Thank you again and again.

Round 2
Reviewer 2 Report
Comments and Suggestions for Authors
The revised manuscript appears much improved. The responses provided addressing my questions and comments seem satisfactory.